# L-Lysine as the Molecule Influencing Selective Brain Activity in Pain-Induced Behavior of Rats

**DOI:** 10.3390/ijms20081899

**Published:** 2019-04-17

**Authors:** Liudmila A. Severyanova, Victor A. Lazarenko, Dmitry V. Plotnikov, Maxim E. Dolgintsev, Alexey A. Kriukov

**Affiliations:** 1Pathophysiology Department, Kursk State Medical University, Kursk 305041, Russia; makdol@mail.ru (M.E.D.); fluri@yandex.ru (A.A.K.); 2Department of Surgical Diseases FPE, Kursk State Medical University, Kursk 305041, Russia; kurskmed@mail.ru; 3Psychiatry Department, Kursk State Medical University, Kursk 305041, Russia; plot1967@yandex.ru

**Keywords:** L-lysine, pain-induced behavior, aggression, selectivity of brain activation

## Abstract

Lysine-rich proteins are some of the most important proteins of neurons and it has become necessary to investigate the possible role of L-lysine as a brain functioning regulator. The purpose of our study is to identify the characteristics and the mechanisms of L-lysine effects on the different types of pain-induced behavior in the stimulation of tail and foot-shock models in 210 adult male Wistar rats. L-lysine was administered in intraperitoneal or intracerebroventricular injections in doses of 0.15–50.0 µg/kg. When a tail is irritated, L-lysine was found to enhance pain sensitivity and affective defense after both intraperitoneal and intracerebroventricular administration. In the case of unavoidable painful irritation of a pair of rats with both types of L-lysine administration, there was no direct correlation of the severity of pain with defensive reactions and outbursts of aggression. This indicates a more complex integration of the activity of brain structures in this situation of animal interaction, which was confirmed by the results of the direct amino acid action on the periventricular brain structures. Our findings show that L-lysine influences the selective brain activity in dependence on the biological significance of pain-induced behavior.

## 1. Introduction

Lysine is a diaminomonocarbonic essential acid that is incorporated into almost all the proteins in animals and plants. It is known that the specificity of brain nuclear proteins is expressed to a greater degree than in other organs. In the neurochemical analysis of neuronal nuclear proteins with high lysine content, the specific histone lysine methyltransferases were described. The enzymes were closely connected with chromatin. Thus, it was found that the acetylation of L-lysine residues controlled the activity of histones, highly mobile proteins groups, nuclear and transcription factors. Furthermore, the methylation of lysine specific residues in histones further regulated the chromatin structure and gene expression [1]. These facts suggest that L-lysine itself might have modulatory actions on the cellular physiological processes—cell proliferation, differentiation and excitability as well as intercellular interaction by means of neurotransmitter systems.

An increased content of lysine-rich proteins was found in the evolutionally novel brain structures such as the cerebral cortex, especially in the pyramidal cells [2]. Other brain structures were arranged according to these proteins content as follows: paleocortex, hippocampus, subcortical and brain stem nuclei. It was shown that the qualitative and quantitative characteristics of distinct brain nuclei were manifested in the peculiar emotional and behavioral reactions and learning [2].

A significant cycle of studies on the neurotropic effects of L-lysine was carried out at the end of the 20th and beginning of the 21st centuries. The anxiolytic action of the amino acid was established under stress, in particular in rats on foot-shock and handling models [3]. The output would be based on the effects of a poor L-lysine diet and its addition to food. A decrease in anxiety and sympathetic activity was also found against the background of a rich L-lysine diet, measured by determination of skin galvanic conductivity.

Similar L-lysine activity was observed in studies on humans. In healthy volunteers, a decrease in the high level of personal anxiety has been established under the influence of L-lysine and L-arginine addition in food for several days [4]. Such a diet also reduced the emotional stress associated with public speaking. L-lysine also showed the effects of an antidepressant and anticonvulsant.

Attempts were also made to uncover the mechanisms of the anxiolytic action of L-lysine in particular, a decrease in circadian adrenaline secretion in the hypothalamus was found in a diet rich in L-lysine. It was found that the mechanisms of these amino acid effects include increased affinity of the GABA-benzodiazepine–receptor complex [5]. The role of a neurotransmitter or neuromodulator in the CNS–GABA inhibitory system is assigned to the predominant L-lysine metabolite in the brain tissue, which is named pipecolic acid [6]. L-alpha-aminoadipate is another amino acid metabolite, which was suggested to play a role of neurotransmitter and the weak competitive inhibitor of L-glutamate and L-aspartate intake by the brain cells.

In a few investigations of L-lysine neurotropic action under stress conditions, an anxiolytic effect was found in rats and humans, which was connected with neurotransmitter responses as follows: release of norepinephrine from ventro-medial hypothalamus and the serotonin release in the central amygdala nucleus [7].

The discovery of the neurotropic effects of L-Lysine has increased the relevance of a study of its action on pain sensitivity and mechanisms of pain development. Meanwhile, a pain is a universal response of the human body and higher animals to damage or its threat, on which the survival of species depends. Therefore, it should be considered that L-lysine has an obvious neurotropic effect. The study of its action on pain and pain-induced behavior is of particular relevance. This consideration determined the goal of our work.

In the light of the above analysis, the purpose of our study was to identify the characteristics and the mechanisms of L-lysine effects on the different types of a pain-induced behavior in rats. In order to ensure that the effect of L-lysine occurs within brain structures, we have evaluated the effect of the amino acid after intraperitoneal injection, during which it enters the median eminence of the hypothalamus, as well as after its intracerebroventricular injection, implying a primary action on the periventricular structures. Pain-induced behaviors were studied using the model of electrical tail stimulation with painful behavioral reactions and a foot-shock one triggering the greater complex defensive-aggressive behavior.

## 2. Results

The results obtained on the model of electrical stimulation of a tail after intraperitoneal administration of L-lysine are represented in Figure 1. The average values of the thresholds for the occurrence of pain manifestations (a tail tension and head turning in the direction of stimulus) and emotional-affective defense (vocalization, body movements and electrodes biting) are given. As can be seen from Figure 1, L-lysine administration in the used doses produced algesic effects as revealed by a decrease of the threshold values of all painful and behavioral reactions. The action of the amino acid was greater with an increase in the applied dose. This especially concerns the enhanced pain-induced affective defense. 

The obtained decrease in the threshold values seemed to be statistically significant according to ANOVA and the post hoc Dunnett test as follows: tension of a tail—F(6.127) = 7.9933, *p* = 0.0000; head turn—F(6.124) = 7.2128, *p* = 0.0000; a body rotation—F(6.124) = 7.8159, *p* = 0.0000; a biting of electrodes—F(6.30) = 7.94916, *p* = 0.0000. The maximum effects were observed after the administration of L-lysine at a dose of 50 μg/kg when the increase in threshold values reached 20%–42%.

We discovered the same effects after the intracerebroventricular administration of L-Lysine. The Figure 2 illustrates the obtained results as the decrease in the threshold values that is confirmed by the statistical significance namely ANOVA: for a tail tension—F (3.41) = 23.9603, *p* = 0.0000; a head turn—F(3.43) = 3.9414, *p* = 0.01433; a vocalization—F(3.47) = 5.9122, *p* = 0.0016; a body rotation—F(3.41) = 3.95378, *p* = 0.01448; an electrode biting—F(3.37) = 4.0162, *p* = 0.01431. The dose of 15 μg/kg showed the most significant effects on the painful and affective response.

In the next set of experiments, we tested the effects of L-lysine on the aggressive-defensive behavior reproduced in the foot-shock model. The obtained results are represented in Figure 3.

It was revealed that after intraperitoneal administration, the influence of an amino acid depended on an administered dose. With the injection of L-lysine in the doses of 1.5; 5.0 and 15.0 µg/kg the threshold values of the nociceptive reactions (flinching, vocalization) decreased up to 18% and the defensive running decreased to 13% in relation to the control values. As for the aggression response, its increase was manifested in the greater number fighting reactions by 20%–50 % in comparison with the reference value.

The administration of an amino acid in the doses 0.5 and 50.0 μg/kg induced the opposite effects as follows: antinociception (flinching—ANOVA: F(6.82) = 3.076844, *p* = 0.0091); decreased avoidance—ANOVA: F(6.115) = 11.1984, *p* = 0.0000 and antiaggressogenic effect revealed by the decrease in the fighting rate (up to 10% of the total number of samples).

As for the effects of intracerebroventricular L-lysine administration to rats in the foot-shock model (Figure 4), the administration of both doses was accompanied with a decrease in pain sensitivity that did not reach a significant level. However, on reaching the threshold value of the arising of attacks, the fighting rate turned out to be higher than the reference value.

## 3. Discussion

First of all, it is necessary to emphasize that we investigated the forms of behavior induced by a painful stimulation that was similar to the physical characteristics but had different biological significance.

Thus, after electrocutaneous irritation of a tail the painful spinal reflex—the tail tension—was more sensitive to the action of L-lysine. With an increase in the dose a defensive behavior appears to become more pronounced and is revealed by “attack” on a physical object—biting of electrodes. The source of pain can be considered as an object of affective defense. With an increase in the dose of an amino acid, the supraspinal structures which are involved in the formation of defensive behavior become more highly activated.

After pain stimulation of the paws on an electrified floor, i.e., under the conditions of possible communication and perception of visual and odor stimuli, painful and defensive reactions simultaneously arise in two animals. The behavioral response ends in cumulative outbursts of aggression in the form of unfolded species-typical attacks directed at a partner because of an unavoidable irritation. In this situation, the effect of L-lysine was different depending on whether the dose was activating or inhibitory.

It was postulated that an adverse stimulus was encountered, which automatically results in a negative affect that triggers escape-related and aggression-related responses [8,9]. In addition, both operant and respondent aggression can occur concurrently and thus, defensive behavior in the foot-shock model is more complex as it involves the interaction of many brain structures. Actually, the cortical areas, amygdala, hypothalamus and the midbrain periaqueductal gray matter interconnected by several pathways constitute a longitudinally organized neuronal system that is responsible for the defensive behavior [10]. It was shown that the septo-hippocampal system and amygdala could detect threatening stimuli and the degree of threat, with the periaqueductal gray being part of a final common pathway of affective defense [10]. In several studies the involvement of the hippocampal formation in nociception was suggested [11]. The defense is associated with the emotional experience of fear. It is known that these brain structures also serve as the substrate for an aggressive impulse and the ultimate escalation to aggressive behavior [12].

The methods of intraperitoneal and intracerebroventricular administration of L-lysine ensure its primary access to various areas of the brain and allow us to determine importance for the formation of amino acid effects. When administered intraperitoneally, L-lysine could affect the hypothalamus since the median eminence is a zone of increased permeability in the hematoencephalic barrier. The injection of L-lysine into the cerebral ventricle provokes its direct effect on the periventricular structures, which increases the emotional response to painful stimulation. It is known that the increased (3H-labeled) lysine absorption and incorporation into brain mice proteins were observed under the influence of the electrical pain stimulation [13].

Analysis of the results of our study showed the dependence of the L-lysine effects on the biological significance of pain-induced behavior. Thus, after pain stimulation of a tail, the predominant effect of the amino acid was an increase in pain sensitivity that was most pronounced with intraperitoneal administration, i.e., an algesic effect. The emotional response and manifestations of affective defense and thus, the involvement of limbic structures was also pronounced. These data are consistent with the results of intracerebroventricular amino acid administration which demonstrated enhanced affective defense.

In the case of painful stimulation of rat pairs, the algesic effect of L-lysine after intraperitoneal administration was not pronounced but even in its absence, there was a greater level of aggression compared to the control. The analgesic and antiaggressogenic effects of the amino acid only coexisted at a dose of 50 µg/kg. After the central injection of L-lysine, the aggressogenic effect was not associated with a change of pain sensitivity. It is possible to conclude that the supraspinal integrative interaction played a prevalent role in this administration due to the primary activation of the periventricular system.

The mechanisms of the neurotropic effects of L-lysine might include its interaction with the neurotransmitter systems of the brain, due to which the integration of defensive behavior and aggression takes place. Thus, it was established that the foot-shock procedure provokes the release of norepinephrine in the amygdala [14] and promotes an aggressive response. An increase in the brain content of norepinephrine was synchronized with an increase in post-stress anxiety [15]. In contrast, the brain serotoninergic system, including the medial prefrontal cortex, plays an important role in the controlling of acquisition, manifestation and extinction of fear [10,15,16,17]. The cholinergic, opioidergic and GABA-ergic systems of the dorsal hippocampus seemed to be involved in the modulation of nociception in guinea pigs [18]. It was shown that serotonin- and dopamine-receptors are located on the arcuate nucleus neurons of the hypothalamus, which take part in the defensive and aggressive reactions [19,20]. Furthermore, the phenomenon of the coexistence and corelease of more than one transmitter in neurons was confirmed as the corelease of dopamine and glutamate from the midbrain neurons [21,22]. It is natural to assume that the effects of L-lysine can be accomplished through a change in the activity and interaction of the neurotransmitter systems with a specific neuroanatomical localization. This assumption is confirmed by the data that when there is a deficiency of L-lysine in rats, the release of serotonin in the central nucleus of the amygdala [23] and norepinephrine in the ventro-medial hypothalamus decreases [24]. In addition, the amino acid could manifest its properties as a competitive serotonin antagonist, which were established in vitro on the ileum of the guinea pig [23]. Thus, L-lysine could influence brain structures, which play an important role in the formation of agonistic behavior.

## 4. Materials and Methods 

### 4.1. Animals

The study was performed on 210 adult male Wistar rats weighing 180–200 g that were kept in standard conditions: free access to food and water, temperature of 22 ± 2 °C and light/dark cycle of 12 h. In the experiment, they were divided into groups, which had 10 animals each. Behavioral patterns of pain-induced behavior were investigated.

In our study, all animal handling and testing was performed in accordance with the Council Directive 2010/63EU of the European Parliament and the Council of 22 September 2010 on the protection of animals used for scientific purposes and the Local University Committee (Protocol N2, 05.11.2013). Each animal was used in the experiment only once and each irritation time was strictly limited in its duration and intensity, which was immediately stopped once there was aggression.

### 4.2. Behavioral Testing

1. The first model involved electrodermal irritation of the tail, in which lamellar electrodes were applied to the base of the tail and the thresholds (in mA) of consequently developing reactions to pain stimulation were recorded. At the same time, the level of pain sensitivity (reaction of tail tension and head rotation to electrodes) and the thresholds of components of emotional–affective behavior (vocalization, rotation, biting of electrodes) were evaluated. The experiments were performed under the conditions of free behavior of an animal in a chamber, which allowed a researcher to observe these experiments without affecting the behavior of a rat. The maximum duration of the stimulation was about 3 min (gradually 0.01 mA/s).

2. The second model was aggressive–defensive behavior in the case of inescapable electrical stimulation of the paws. Rats were placed in pairs in a chamber with an electrified wire floor and stimulated with gradually increasing (1 V/s) the current generated by a programmable stimulator. Both rats of the test pair had colored marks on their hair. The thresholds of successively developing components of behavior (flinching, vocalization, rising up, running, fighting) and the number of attacks (in% of the number of tests) were measured within certain voltage limits (70 V). Irritation was stopped when an aggressive reaction occurred or when this limit was reached. The maximum duration of the stimulation is about 70 s. Two trials were carried out at an interval of 1 min. Thus, two stimulation tests were carried out on each of ten rats in pairs. The total number of samples for each behavioral reaction—to 20. In both models the experiments were conducted by two researchers, combining observation and automatic registration. At the same time, the appearance of each behavioral component was recorded on an apparatus (multimeter) and the threshold of its appearance—with a software electrostimulator. The researchers easily heard voice reactions of each rat separately. All the indices were immediately recorded in a protocol.

In each series of studies, the rats were separated into control and experimental groups. They were simultaneously obtained from the nursery in the Russian Academy of Sciences. L-lysine was administered in intraperitoneal (140 rats) or intracerebroventricular (70 rats) injections.

### 4.3. L-lysine

L-lysine (ICN) was dissolved in saline and injected intraperitoneally 12 min before the experiment in doses of 0.15, 0.5, 1.5, 5.0, 15.0 and 50.0 mcg (µg) per 1 kg of body weight. The control group of rats received injections of saline in an equivalent volume (1 mL/kg). For the study with intracerebroventricular injections seven days before the beginning of the experiment, rats were implanted with a cannula in the lateral ventricle of the brain under hexenal anesthesia using the stereotaxic atlas Pellegrino L.J., 1981 (Coordinates: A.P.5.5;L.1.6;H.4.0). The amino acid was slowly injected with a microdoser at an appropriate dose (5 or 50 µg) in 3 µL of saline 12 min before an experiment. The fluid flow rate is about 3 µL/s. The control rats received only saline. In terms of the selection of L-lysine doses, the minimum dose of the amino acid was close to its content in the blood and CSF, with each subsequent dose increased by three-fold.

### 4.4. Design

Overall, we have to perform four tasks and accordingly, we designed the following sections to study the effects of L-lysine on the indices of a pain-induced behavior as follows: (1) on the aggressive–defensive behavior (the foot-shock model) after (1a) intraperitoneal and (1b) intracerebroventricular administration; (2) behavioral response in the tail-flick model after (2a) intraperitoneal and (2b) intracerebroventricular administration.

### 4.5. Statistical Analysis

All obtained data are presented as mean +S.E.M. The Shapiro–Wilk test was applied to verify normality. A one-way ANOVA was used to compare saline and treated groups of rats and was followed by the post hoc Dunnett’s test. The level of significance is at 95%, *p* < 0.05. The statistical software was Analyse it. Normal edition. Version 5.11.3 (Leeds, UK).

## 5. Conclusions

The obtained data showed that L-lysine administration influenced the pain sensitivity and the central mechanisms of a pain-induced behavior in both studied models. The effects of amino acid in a certain way depended on a route of administration, dose values and the behaviors. L-lysine administration in the model of a tail stimulation increased a nociceptive spinal reflex (a tail tension) and affective defense after both intraperitoneal and intracerebroventricular injections. Inescapable painful irritation of paired rats (foot-shock) induced a greater complex defensive-aggressive behavior. The effects were preferentially reduced by the intraperitoneal administration of L-lysine. All together, the obtained findings showed that L-lysine effects on nociception and pain-induced behavior depended on the biological significance of the situation of a painful stimulation.

## Figures and Tables

**Figure 1 ijms-20-01899-f001:**
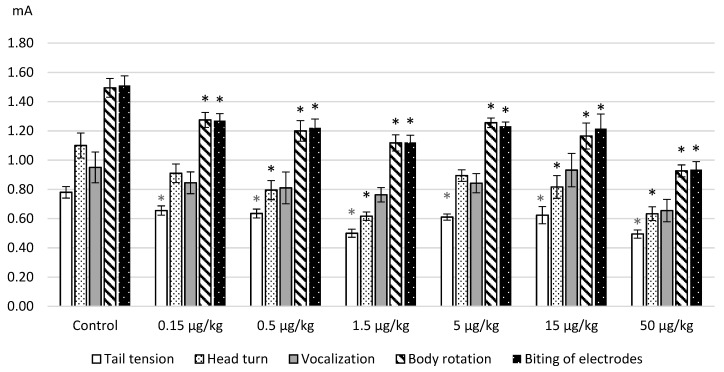
The thresholds (mA, M ± m) of the painful and affective reactions of rats in the tail electric stimulation test after intraperitoneal L-lysine administration. Note: the obtained data were analyzed by ANOVA and the post hoc Dunnett’s test; * *p* < 0.05–0.001 in comparison with the control values.

**Figure 2 ijms-20-01899-f002:**
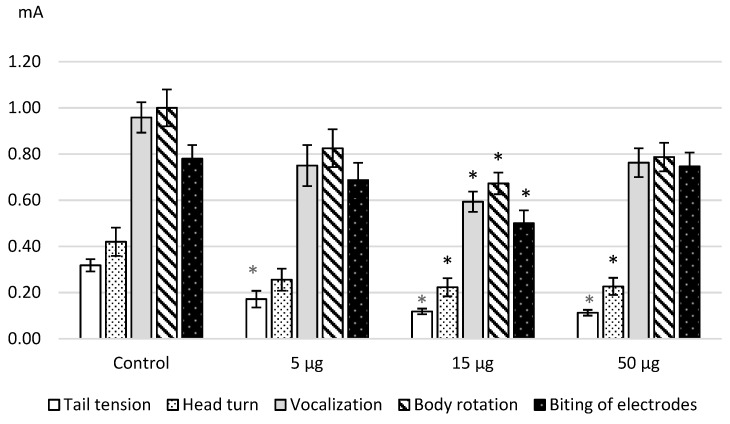
The thresholds (mA, M ± m) of the painful and affective reactions of rats in the tail electric stimulation test after intracerebroventricular L-lysine administration. Note: the obtained data were analyzed by ANOVA and the post hoc Dunnett’s test; * *p* < 0.05–0.001 in comparison with the control values.

**Figure 3 ijms-20-01899-f003:**
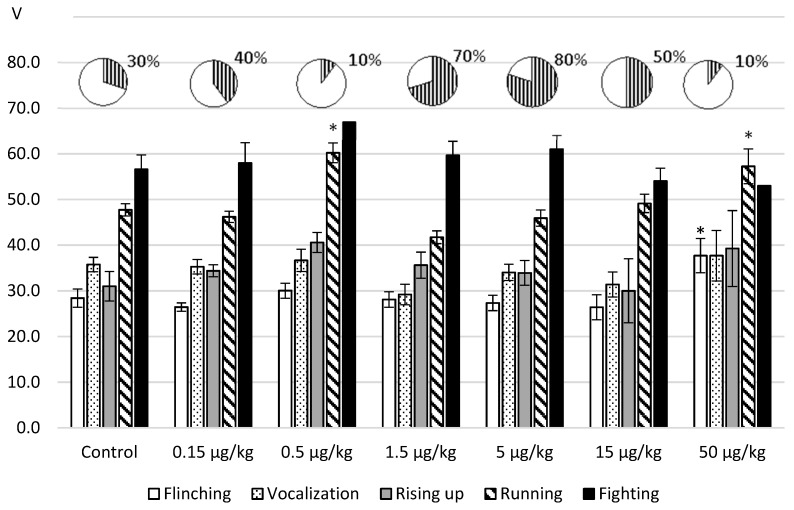
The threshold values (V, M ± m) of the aggressive–defensive behavior of rats in foot-shock test after intraperitoneal administration. Note: the obtained data were analyzed by ANOVA and the post hoc Dunnett test; the round diagrams: the fighting rate (shaded) as a percentage of the total number of samples: two stimulation tests were carried out on each of ten rats in pairs (all samples—to 20).* *p* < 0.05–0.001 in comparison with the control values.

**Figure 4 ijms-20-01899-f004:**
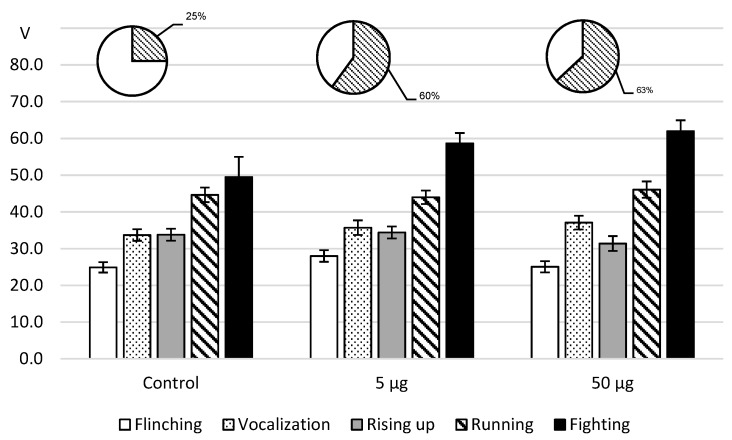
The thresholds values of rat aggressive-defensive behavior (V, M ± m) in the foot-shock test after intracerebroventricular L-lysine administration. Note: the obtained data were analyzed by ANOVA and the post hoc Dunnett test; the round diagrams: the fighting rate (shaded) as a percentage of the total number of samples: two stimulation tests were carried out on each of ten rats in pairs (all samples—to 20).

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
