# Peer review of "L-Lysine as the Molecule Influencing Selective Brain Activity in Pain-Induced Behavior of Rats"

_ijms, 2019, doi:10.3390/ijms20081899_

Round 1

Reviewer 1 Report

The article by Severyanova et al reports the effect of L-Lysine on pain-induced behavior in wistar rats. Two models were used: the tail-flick and the foot-shock tests. The authors looked at the behavior in each test in rats receiving either saline (vehicle of L-Lysine) or different concentrations (0.15 to 50.0 μg/kg) of L-Lysine administered in the ventricle or intraperitoneally. The authors conclude that L-Lysine has an impact on the behavioral responses triggered by pain and suggest that the responses are dependent on the biological significance of pain-induced behavior.

The article has been improved. The idea to transform the tables in figures is appreciated. The statistics correspond to the experimental design. The discussion has been ameliorated. There are still some points that should be clarified.

Major issues

1. The report of the results of the ANOVA should indicate [one-way ANOVA F(2,27) = 1.397, p = .15)] (only an example unrelated to the data of the authors). I don’t understand the reason why there is two results for the same ANOVA that are reported in the results describing figure 1 (page 2, end of the page). Normally, the tail tension should include all the seven groups (the results obtained in controls and in treated rats – 6 doses). Therefore, the ANOVA result is single for one parameter and should be something like F(6,..)=…, p<0….  The authors have chosen the Tukey’s test as post hoc (methods). Maybe they could use a less conservative test (the Tukey's test might limit the occurence of significant difference which could give type 2 statistical errors) such as the protected least significance difference for instance due to the nature of the data.

2. The report of the post hoc test in the figure (but maybe they report a sort of ANOVA) is curious and the text doesn’t match what is reported in the figure. Regarding the former aspect, look at figure 1, tail tension parameter. The authors report significant differences for the two higher doses and not for the dose of 1.5 µg/kg (looking at those data, I don't see a good reason why 0.15 µg/kg won't be significant). I also raise some doubt about the significance at 0.5 µg/kg and the absence of significance at 50 µg/kg when I look at the vocalizations data. The text doesn’t match what is reported in the figure 1 because I implicitly understand that the ANOVA was not significant for head turn, body rotation or biting electrode.

The remarks made for figure 1 are true for figures 2 and 3.

3. The distribution (round diagrams) should be explained in the methods (at least their calculation).

4. In the light of the results of statistical analysis, some changes could be envisioned in the results and the discussion.

5. The legend to figures should report the name of the statistical test corresponding to the symbols

Author Response

Reviewer’s comments – responses

The article has been improved. The idea to transform the tables in figures is appreciated. The statistics correspond to the experimental design. The discussion has been ameliorated. There are still some points that should be clarified.

1. The report of the results of the ANOVA should indicate [one-way ANOVA F(2,27) = 1.397, p = .15)] (only an example unrelated to the data of the authors). I don’t understand the reason why there is two results for the same ANOVA that are reported in the results describing figure 1 (page 2, end of the page). Normally, the tail tension should include all the seven groups (the results obtained in controls and in treated rats – 6 doses). Therefore, the ANOVA result is single for one parameter and should be something like F(6,..)=…, p<0….  The authors have chosen the Tukey’s test as post hoc (methods). Maybe they could use a less conservative test (the Tukey's test might limit the occurence of significant difference which could give type 2 statistical errors) such as the protected least significance difference for instance due to the nature of the data.

2. The report of the post hoc test in the figure (but maybe they report a sort of ANOVA) is curious and the text doesn’t match what is reported in the figure. Regarding the former aspect, look at figure 1, tail tension parameter. The authors report significant differences for the two higher doses and not for the dose of 1.5 µg/kg (looking at those data, I don't see a good reason why 0.15 µg/kg won't be significant). I also raise some doubt about the significance at 0.5 µg/kg and the absence of significance at 50 µg/kg when I look at the vocalizations data. The text doesn’t match what is reported in the figure 1 because I implicitly understand that the ANOVA was not significant for head turn, body rotation or biting electrode. The remarks made for figure 1 are true for figures 2 and 3.

Response 1, 2: We changed.

All obtained data are presented as mean +S.E.M. The Shapiro – Wilk test was applied to verify normality. A one-way ANOVA was used to compare saline and treated groups of rats and was followed by the post hoc Dunnett test. The level of significance is at 95%, P<0.05. The statistical software was Analyse it. Normal edition. Version 5.11.3.

3. The distribution (round diagrams) should be explained in the methods (at least their calculation).

4. In the light of the results of statistical analysis, some changes could be envisioned in the results and the discussion.

5. The legend to figures should report the name of the statistical test corresponding to the symbols

Response 3, 4, 5: The necessary correction was made.

Reviewer 2 Report

The manuscript has been significantly improved.

Author Response

The manuscript has been improved.

Round 2

Reviewer 1 Report

This new version is much more attractive than the previous ones. The use of the Dunnett’s test is perfect. I have only minor comments left as follows:

-         lines 102 and 103: I’m not sure about the F(2,37) or F(2,43) as the number of groups is 4 and I’m expecting F(3,…).

-         There are modifications of the number of observations in the ANOVA which is likely due to loss of data. The authors are welcome to briefly mention this (accidental manipulation, problems of recording, outliers) in the statistical analysis of the data (Methods).

-         Figure 3: two errors bars are not shown.

-         Discussion and methods: the authors identified several paragraphs. Some of them could be combined, notably those composed by only one sentence and continuing the same idea.

-         Conclusions, Lines 286-289: “a certain tendency….. seemed to be…”. Please remove these two sentences and indicate that “the effects were preferentially reduced by the intraperitoneal administration of lysine”.

-         Lines 191-192: please rephrase “After the central injection of L-lysine, the aggressogenic effect was observed on the background not changed pain sensitivity”

Author Response

Reviewer’s comments – responses

This new version is much more attractive than the previous ones. The use of the Dunnett’s test is perfect. I have only minor comments left as follows:

1) lines 102 and 103: I’m not sure about the F(2,37) or F(2,43) as the number of groups is 4 and I’m expecting F(3,…)

Response 1. We are so sorry for the mistake made by the operator on registering the values of two factors. ANOVA is a new program for us, and, unfortunately, the errors were not noticed in time. We are very grateful for the attentive and kind work with our manuscript.

2) There are modifications of the number of observations in the ANOVA which is likely due to loss of data. The authors are welcome to briefly mention this (accidental manipulation, problems of recording, outliers) in the statistical analysis of the data (Methods)

Response 2. Hereinafter, we present only statistically reliable data obtained using ANOVA in order not to overload the text with unreliable reports. In the Figures, the obtained data are presented in full: all the sets of experiments and investigated groups of animals.

3) Figure 3: two errors bars are not shown

Response 3. In Figure 3, two errors bars are not shown for the threshold values of the fighting occurrence after L-lysine administration in doses of 0.5 and 50.0 micrograms. This is due to the fact that attacks originated in one pair of rats.

4) Discussion and methods: the authors identified several paragraphs. Some of them could be combined, notably those composed by only one sentence and continuing the same idea

Response 4. We have combined some paragraphs in the cases witch we revealed.

5) Conclusions, Lines 286-289: “a certain tendency….. seemed to be…”. Please remove these two sentences and indicate that “the effects were preferentially reduced by the intraperitoneal administration of lysine”

Response 5. The correction was made.

6) Lines 191-192: please rephrase “After the central injection of L-lysine, the aggressogenic effect was observed on the background not changed pain sensitivity”

Response 5. The correction was made.

We would like to inform that English language of manuscript was improved.

This manuscript is a resubmission of an earlier submission. The following is a list of the peer review reports and author responses from that submission.

Round 1

Reviewer 1 Report

The article by Severyanova et al reports the effect of L-Lysine on pain-induced behavior in wistar rats. Two models were used: the tail-flick and the foot-shock tests. The authors looked at the behavior in each test in rats receiving either saline (vehicle of L-Lysine) or different concentrations (0.15 to 50.0 μg/kg) of L-Lysine administered in the ventricle or intraperitoneally. Whatever the mode of administration, they report that L-Lysine enhanced pain sensitivity and affective defense when the tail is irritated. In the case of the foot-shock models where rats were paired, there was no clear relationship for the severity of pain and defensive reactions but there was outburst of aggression between congeners. The authors conclude that L-Lysine has an impact on the plasticity of the behavioral responses triggered by pain and suggest that the responses are dependent on the biological significance of pain-induced behavior.

The article completes previous data showing that L-Lysine interferes on behaviors. As indicated by the authors, it was reported for anxiety and the authors report now arguments for its impact on pain-induced behaviors. L-Lysine could appear as an alert system. The paper could be interesting but there are numerous points that should be ameliorated before considering its publication.

Major points:

1. The reason why L-Lysine has been administered both icv and ip has not been explained. It should have been explained in the introduction and better delineated in the methods and/or the results. As a reader, my question is: “are the authors expecting some differences between the two modes of administration and why?” We can read in the discussion line 135 (too late by the way): “Due to the use of two kinds of L-lysine administration we had the opportunity to “change” - (what does it mean?) - the brain structures that were initially available for its action”. This statement which needs rephrasing is quite speculative and the following text too.

2. As a consequence, the paper is not well written. The above remark is one aspect of the problem (the authors have to focus on what they are actually presenting). Another aspect is the discussion which is a little bit outside of the scope of the study (example: discussion on neurotransmitter systems, corelease of neurotransmitters etc which are pointless here). Finally, there is a problem with the English language including grammatical errors, misspellings, sometimes long sentences without a verb. I had sometimes difficulties to follow the authors. For instance: line 141: “It is known that the increased (3H) absorption and incorporation into brain mice proteins under influence of the electrical foot stimulation [10].”

3. The statistical treatment of the data is incorrect from my point of view. If I understood correctly, the authors performed multiple pairwise comparisons using Student’s t-test. Since they are numerous groups, it means that the same group has been compared several times with other groups, generating an error in the statistical analysis even larger than the power of the test itself. In this situation, the authors have to use a one-way ANOVA to determine if there are statistical differences between all the groups (saline to 50µg/kg) (the ANOVA compensating the problem generated by the t-test). In case of significance, the authors have to use a post hoc test to determine where the differences are; of course, the post hoc test is a test specifically designed to handle multiple comparisons between groups and it cannot be a t-test.

4. Additional attention has to be given to the notion of biological significance or plasticity. The plasticity is unclear and I see a classical acute neuropharmacological response to a compound, here L-Lysine.

5. The methods are not sufficiently described. Indeed, the numerous behavioral parameters have been mentioned but the way they were measured has not been reported. For instance, how the authors could measure vocalizations? This is true for all of them. The duration of the behavioral evaluation is unclear as well. These precisions are mandatory.

Minor points

The authors could pay attention to English meaning of some words: character (which has been wrongly employed; feature or characteristic could be better; base should be replaced by basis; line 124: adverse; line 130, have could better than has)

The authors could pay also attention to the way they write a sentence. For instance, line 130, they quoted: “The data obtained has (have) shown that the septo-hippocampal system and amygdala could detect threatening….” It is written in a way that it suggests the authors did the experiments, which is not the case.

Methods, line 204: “the amino acid was slowly administered with….”: what is a microdoser (is it a perfusion pump (from where)? They could report the flow rate as well.

Author Response

Dear colleague,

We are very grateful for your analysis of our manuscript and thoughtful comments.

Point 1: The reason why L-Lysine has been administered both icv and ip has not been explained. It should have been explained in the introduction and better delineated in the methods and/or the results. As a reader, my question is: “are the authors expecting some differences between the two modes of administration and why?” We can read in the discussion line 135 (too late by the way): “Due to the use of two kinds of L-lysine administration we had the opportunity to “change” - (what does it mean?) - the brain structures that were initially available for its action”. This statement which needs rephrasing is quite speculative and the following text too.

Response 1: The corrected statement (line 135).

The methods of intraperitoneal and intracerebroventricular administration of L-lysine ensure its primary access to various areas of the brain and allow us to determine importance for the formation of amino acid effects.

Point 2: As a consequence, the paper is not well written. The above remark is one aspect of the problem (the authors have to focus on what they are actually presenting). Another aspect is the discussion which is a little bit outside of the scope of the study (example: discussion on neurotransmitter systems, corelease of neurotransmitters etc. which are pointless here). Finally, there is a problem with the English language including grammatical errors, misspellings, sometimes long sentences without a verb. I had sometimes difficulties to follow the authors. For instance: line 141: “It is known that the increased (3H) absorption and incorporation into brain mice proteins under influence of the electrical foot stimulation [10].”

Response 2: The manuscript contains the latest data on the organization of the neurotransmitter systems of the brain to illustrate the complex neurochemistry of the brain and show the prospects for studying the effect of L-lysine as an active modulator.

Corrected sentence (line 141): It is known that the increased (3H-labeled) lysine absorption and incorporation into brain mice proteins were observed under the influence of the electrical pain stimulation [10].

Thank you very much for the advice concerning English editing our manuscript. The Editing Office has carried out the necessary correction.

Point 3: The statistical treatment of the data is incorrect from my point of view. If I understood correctly, the authors performed multiple pairwise comparisons using Student’s t-test. Since they are numerous groups, it means that the same group has been compared several times with other groups, generating an error in the statistical analysis even larger than the power of the test itself. In this situation, the authors have to use a one-way ANOVA to determine if there are statistical differences between all the groups (saline to 50µg/kg) (the ANOVA compensating the problem generated by the t-test). In case of significance, the authors have to use a post hoc test to determine where the differences are; of course, the post hoc test is a test specifically designed to handle multiple comparisons between groups and it cannot be a t-test.

Response 3: Overall, we have to perform four tasks and accordingly, we designed the following sections to study the effects of L-lysine on the indices of a pain-induced behavior as follows:

1) on the aggressive–defensive behavior (the foot-shock model) after 1a) intraperitoneal and 1b) intracerebroventricular administration;

2) behavioral response in the tail-flick model after 2a) intraperitoneal and 2b) intracerebroventricular administration.

We didn’t compare three or more group means for statistical significance (saline to 50 µg/kg). In each section of our work, we compared the threshold values of animal behavior only between group that was subjected to L-lysine injection in precise dose and control group. Therefore, we considered a pairwise comparison with the use of the t-test to be acceptable according to the task. If such a task is set forth, we will use the valuable advice to apply ANOVA.

Point 4: Additional attention has to be given to the notion of biological significance or plasticity. The plasticity is unclear and I see a classical acute neuropharmacological response to a compound, here L-Lysine.

Response 4: The formation of different behavioral responses to a similar pain irritation of the skin, but in biologically different situations is the result of selective activation of brain structures and, therefore, proves plasticity in brain activity.

L-lysine effects were observed including the minimum doses close to the physiological values and may be estimated not as exclusively pharmacological responses.

Point 5: The methods are not sufficiently described. Indeed, the numerous behavioral parameters have been mentioned but the way they were measured has not been reported. For instance, how the authors could measure vocalizations? This is true for all of them. The duration of the behavioral evaluation is unclear as well. These precisions are mandatory.

Response 5: The consequently arising components of a pain-induced behavior, including vocalization, were estimated by the threshold value of the stimulation when a component arised. The maximum duration of the stimulation in tail-flick model was about 3 min (gradually 0,01mA/sec). The duration of behavioral response in foot-shock model is about 70 sec (gradually 1V/sec).

Minor points: The authors could pay attention to English meaning of some words: character (which has been wrongly employed; feature or characteristic could be better; base should be replaced by basis; line 124: adverse; line 130, have could better than has)

The authors could pay also attention to the way they write a sentence. For instance, line 130, they quoted: “The data obtained has (have) shown that the septo-hippocampal system and amygdala could detect threatening….” It is written in a way that it suggests the authors did the experiments, which is not the case.

Methods, line 204: “the amino acid was slowly administered with….”: what is a microdoser (is it a perfusion pump (from where)? They could report the flow rate as well.

Response Minor points:

The lines 124,130 were ccorrected. The line 204. A microdoser is an apparatus for solution injection not the perfusion pump. The fluid flow rate is about 3-4 sec.

Reviewer 2 Report

This manuscript by Severyanova et al describes the putative role of l-lysine as neuromodulator in pain behavior. The research design was rigorous, and the statistical analyses used were appropriate to answer the research question. These finding suggest an interesting but complex phenomenon of brain plasticity and behavior. 

Minor changes required

1.     Mention what are the different rationales for IP and ICV, lysine dose selection either in result or discussion?

2.     Line 130-132: Data from current manuscript doesn’t support this statement “The data obtained has shown that……. of affective defense”.  Either provide reference to this statement or omit the sentence.

Author Response

Thank you very much for your thoughtful analysis of our manuscript.

Point 1:  Mention what are the different rationales for IP and ICV, lysine dose selection either in result or discussion?

Response 1: The methods of intraperitoneal and intracerebroventricular administration of L-lysine ensure its primary access to various areas of the brain and allow us to determine importance for the formation of amino acid effects.

When choosing doses, we proceeded from the task of studying to a greater extent the physiological and not pharmacological effects of L-lysine. Therefore, the minimum dose of the amino acid was close to its content in the blood and CSF, and each subsequent dose was increased three times in accordance with the known rule of drug-effects assessment.

Point 2:  Line 130-132: Data from current manuscript doesn’t support this statement “The data obtained has shown that……. of affective defense”.  Either provide reference to this statement or omit the sentence.

Response 2: Lines 130-132 were corrected.

Round 2

Reviewer 1 Report

The revised version is not satisfactory. The english has been improved but not the scientific background.

Point 1: the authors have to present the rational to compare the effect of Lysine when administered icv and ip. It has to be explained in the introduction in the text, not only for me. In the discussion, this is too late, although it has to be discussed as well.

Point 2: not sufficiently considered because some parts are still completely speculative (still line 351 and beyond).

Point 3: Indeed, the authors have to perform the ANOVAs (parametric or non parametric depending on the nature on the parameters). Here, they compared several times the same group with t-tests: this procedure is totally wrong.

Point 4: I still don't see the plasticity in these experiments. If the authors prefer, I see only a classical physiological response, not necessarily implying plasticity.

Point 5: I still don't know how vocalizations or other parameters were measured.

Author Response

Response to reviewer 1 Comments

Dear colleague,

We are very grateful for your attention to us.

Point 1: the authors have to present the rational to compare the effect of Lysine when administered icv and ip. It has to be explained in the introduction in the text, not only for me. In the discussion, this is too late, although it has to be discussed as well.

Response 1: We believe that in order to achieve this goal it is important to solve the following tasks.

1. To ensure the primary effect of L-lysine on various brain structures. This problem can be solved through the use of various methods of amino acid injection in particular, intraperitoneal, in which it primarily enters the medial eminence of the hypothalamus, and intraventricular, providing a primary effect on the periventricular structures. 2. To investigate the effects of amino acids on the pain-induced behavior of various biological significance, which requires selective activation of brain structures. We prefer to reproduce two following models on rats: the tail-flick model with painful behavioral reaction and foot-shock one triggering the greater complex defensive-aggressive behavior.

Point 2: not sufficiently considered because some parts are still completely speculative (still line 351 and beyond).

Response 2: Conclusion (changed)

The obtained data showed that L-lysine affects pain sensitivity and pain-induced behavior, and its action depends on the biologic significance of the pain irritation situation. When the tail is irritated, L-lysine enhances pain sensitivity and affective defense for both intraperitoneal and intracerebroventricular administration. At the same time, with an increase in the strength of irritation, the defensive reactions changed more than the painful ones, which indicates the greater enhanced activation of the corresponding supraspinal structures. In the case of unavoidable painful irritation of paired rats with both types of L-lysine administration, there was no direct correlation between the severity of pain and aggression that demonstrates more complex integrative brain activity in such the situation of animal communication. The conclusion is confirmed by the results of the direct amino acid action on the periventricular brain structures when the fighting rate is high without increased painful components.

In general, the obtained results allow the following conclusions to be made.

1. L-lysine has a modulating effect on pain-induced behavior in dependence on its biological type. 2. The mechanisms of this modulating effect include the selectivity of the activation of brain structures.

Point 3: Indeed, the authors have to perform the ANOVAs (parametric or non parametric depending on the nature on the parameters). Here, they compared several times the same group with t-tests: this procedure is totally wrong.

Response 3: As known, the non-parametric Student t- test is the adequate comparison of mean values of two independent groups of data. We suggest to examine the researches as totality of the separately conducted experiments executed by us. On the example of model of electro-skin irritation of a tail after the intraperitoneal administration of L- lysine it was as follows:

1. Estimation of effect of dose of 0.15 µg/kg by comparison to a control group - a t-test is applicable (2 independent groups);

2. Estimation of effect of dose of 0.5 µg/kg by comparison to a control group - a t-test is applicable (2 independent groups);

3. Estimation of effect of dose of 1.5 µg/kg by comparison to a control group - a t-test is applicable (2 independent groups);

4. Estimation of effect of dose of 5 µg/kg by comparison to a control group - a t-test is applicable (2 independent groups);

5. Estimation of effect of dose of 15 µg/kg by comparison to a control group - a t-test is applicable (2 independent groups);

6. Estimation of effect of dose of 50 µg/kg by comparison to a control group - a t-test is applicable (2 independent groups.

We pay attention, that after the above-described processing of data we did not produce comparison of effects between doses, because research was not pharmacological and exposure most or the least effective dose of amino acid was not our aim. By analogical character treatment of the experiments given in all presented series was conducted. This method of processing of data was repeatedly used by us and our colleagues. Data were published in the different magazines including included in the international base of quoting Scopus: DOI: https://doi.org/10.1007/s10517-016-3186-8; https://doi.org/10.1007/s10517-017-3748-4

One - way ANOVA is applicable at plural comparison of data between all the groups (three and more) A classic example can be as following: the comparison of efficiency of influence of preparation 1, preparation 2. preparation 10 under control a placebo on the level of glucose of blood. When the use of t- test is not acceptable for pairwise plural comparisons of data and with the group of placebo and with the groups of the administration of different drugs. If the reviewer can not agree with the above-described explanation we would ask to specify the conсrete references of ANOVA using in studies  with a similar design.

Point 4: I still don't see the plasticity in these experiments. If the authors prefer, I see only a classical physiological response, not necessarily implying plasticity.

Response 4: In our article, we mean functional plasticity as a property of the brain to form a different behavioral response to an identical in physical parameters pain irritation of the skin and its primary nociceptive reaction. In modern neurophysiology, the property of rigidity-plasticity is well known as the ability of the brain to involve mandatory and adaptive elements in the formation of a response. Mandatory brain elements are involved in each answer, and adaptive ones are supplemented depending on certain conditions. We have considered nociceptive pathways as mandatory elements, and the structures responsible for behavioral components are variables adaptive ones.

However, we agree that the term "functional plasticity" of the brain may be replaced by the term "selective brain activation".

Point 5: I still don't know how vocalizations or other parameters were measured.

Response 5: Both rats of the test pair had colored marks on their hair. In both models the experiment was conducted by two researchers, combining observation and automatic registration. At the same time, the appearance of each behavioral component was recorded on an apparatus (multimeter) and the threshold of its appearance - with a software electrostimulator. The researchers easily heard voice reactions of each rat separately. All the indices immediately were recorded in a protocol.

Round 3

Reviewer 1 Report

The article by Severyanova et al reports the effect of ip and icv administration of Lysine at different doses on pain sensitivity and behaviors. This is the second revision.

I have still comments to ameliorate the article.

1) For the point 1: the authors corrections could be as follow: “In order to ensure that the effect of L-lysine occurs within brain structures, we have evaluated the effect of Lysine after its intraperitoneal injection, in which it primarily enters the medial eminence of the hypothalamus, and after its intraventricular injection, implying a primary effect on the periventricular structures. Pain-induced behaviors were studied using the tail-flick model with painful behavioral reaction and foot-shock one triggering the greater complex defensive-aggressive behavior.”

2) For statistics, the procedure adopted by the authors is not correct. What does “independent” mean if the same group is compared several times with other groups? Thus, the authors have to perform a one-Way ANOVA (or Kruskall-Wallis  analysis – non parametric) and then, in case of significance, to use a post hoc test to check where are the eventual differences. If the authors wish to consider only the differences with the control group, it is possible; but in any case, it has to be done using a test handling several groups (ANOVA).

3) line 211: “were immediately recorded”.

4) line 223: fluid flow rate cannot be in seconds.

5) line 254: “in dependence of its biological type”:  What does it mean? It has to rephrased (and simplified…)

6) line 255: please also rephrase this conclusion. I would indicate something like: “the modulatory effects of peripheral L-Lysine administration likely involve brain mechanisms”

7) line 204: what do the authors mean by “flinching” ? Is it an “attack” sign?